# Genome-Wide Characterization and Expression of the *Hsf* Gene Family in *Salvia miltiorrhiza* (Danshen) and the Potential Thermotolerance of *SmHsf1* and *SmHsf7* in Yeast

**DOI:** 10.3390/ijms24108461

**Published:** 2023-05-09

**Authors:** Renjun Qu, Shiwei Wang, Xinxin Wang, Jiaming Peng, Juan Guo, Guanghong Cui, Meilan Chen, Jing Mu, Changjiangsheng Lai, Luqi Huang, Sheng Wang, Ye Shen

**Affiliations:** State Key Laboratory Breeding Base of Dao-di Herbs, National Resource Center for Chinese Materia Medica, China Academy of Chinese Medical Sciences, Beijing 100700, China

**Keywords:** abiotic stress, heat shock factors, *Salvia miltiorrhiza*, SmHsfB2, thermotolerance

## Abstract

*Salvia miltiorrhiza* Bunge (Danshen) is a traditional Chinese herb with significant medicinal value. The yield and quality of Danshen are greatly affected by climatic conditions, in particular high temperatures. Heat shock factors (Hsfs) play important regulatory roles in plant response to heat and other environmental stresses. However, little is currently known about the role played by the *Hsf* gene family in *S. miltiorrhiza*. Here, we identified 35 *SmHsf* genes and classified them into three major groups: SmHsfA (n = 22), SmHsfB (n = 11), and SmHsfC (n = 2) using phylogenetic analysis. The gene structure and protein motifs were relatively conserved within subgroups but diverged among the different groups. The expansion of the *SmHsf* gene family was mainly driven by whole-genome/segmental and dispersed gene duplications. The expression profile of *SmHsfs* in four distinct organs revealed its members (23/35) are predominantly expressed in the root. The expression of a large number of *SmHsfs* was regulated by drought, ultraviolet, heat and exogenous hormones. Notably, the *SmHsf1* and *SmHsf7* genes in SmHsfB2 were the most responsive to heat and are conserved between dicots and monocots. Finally, heterologous expression analysis showed that *SmHsf1* and *SmHsf7* enhance thermotolerance in yeast. Our results provide a solid foundation for further functional investigation of *SmHsfs* in Danshen plants as a response to abiotic stresses.

## 1. Introduction

Sessile plants often suffer from adverse environmental stresses, such as drought, extreme temperature fluctuations, water deficiency, and high salinity [1]. In particular, high temperatures have become a serious concern worldwide due to global warming, which causes yield loss and a decrease in nutrient quality of crop [2]. For instance, it is predicted that for every additional degree Celsius of temperature increase, the global production of wheat, rice, and maize will decrease by 6%, 32%, and 7.4%, respectively [3]. The devastating effects of heat stress are caused by priming protein misfolding, the accumulation of reactive oxygen species and the destruction of cell membranes [4]. To alleviate these adverse effects, plants have developed effective mechanisms of prevention and repair throughout evolution [5]. The transcriptional control of stress-inducible genes enables plant adaption to changing environments [6]. Accordingly, heat-shock factor proteins (Hsfs) are recognized as crucial components of signal transduction chain that mediates the activation of genes responsive to heat and other stresses [7].

Plant Hsfs share a basic modular structure, comprising a DNA-binding domain (DBD), an oligomerization domain (OD), a nuclear localization signal (NLS) and a nuclear export signal (NES). The highly conserved N-terminal DBD specifically binds to heat shock elements (nGAAnnTTCn) in the promoter region of target genes [8]. Hsfs usually exert transcriptional activation dependent on the occurrence of trimerization [9]. The OD contains a hydrophobic heptad repeat (HR-A/B) that forms a coiled-coil motif essential for the oligomerization [10]. In plants, Hsfs can be classified into three subfamilies (A, B or C) based on the characteristics of the HR-A/B regions [11,12]. The transcriptional activation domain (AHA motif) is characteristic of HsfAs [12], while HsfBs contain the conserved tetrapeptide (FLGV) motif that is typically identified as a core repressor domain (RD) [13]. In addition, HSFs contain other well-defined domains, such as the NLS and the NES, which are essential for the dynamic distribution of Hsfs between the cytoplasm and the nucleus [14,15]. The complexity of HSFs in plants suggests a diversification in functional roles in response to different environmental stresses.

Control of thermotolerance is the best characterized function of plant Hsfs. *Hsf1a* is a master regulator of induced thermotolerance in tomato [16]. *HsfA1* (a, b, d and e) and *HsfA2* respond to heat stress (HS) in *Arabidopsis* [17,18,19,20], and HsfA1 and HsfA2 form heterodimers to activate HS-inducible genes [21]. Molecular breeding and genetic engineering are effective and time-saving methods for producing heat-tolerant plants [22]. Hsfs are also involved in plant growth, development and response to other abiotic stresses. The Seed-specific *HsfA9* can regulate embryogenesis and seed maturation in *Helianthus annuus* and *Arabidopsis* [12,23], while quadruple mutants of *AtHsf1s* delay plant growth and increase sensitivity to H_2_O_2_, salt and mannitol [24]. Moreover, the overexpression of *AtHsfB2A* impaired biomass production and gametophyte development in transgenic plants [25], while *HSFA2* enhanced tolerance in combination with HS and high light [19]. A growing number of articles have reported that *HsfA3*, *HsfA4a*, *HsfB2a* and *HsfC1* respond to salt, cold and osmotic induction [9]. In rice, *OsHSFA3* improved drought tolerance by increasing the amount of ABA and polyamine [26] Crucially, the involvement of *Hsfs* in the response of *Salvia miltiorrhiza* to various environmental stresses remains unclear.

*Salvia miltiorrhiza* Bunge is a perennial and widely cultivated herb plant of the *Salvia* genus and *Lamiaceae* family [27]. The dried root of *S. miltiorrhiza* is known as Danshen and has been extensively used in the treatment of cardiovascular and cerebrovascular disease [28]. During growth, unfavorable environmental conditions seriously affect yield and the quality of the Danshen [24], whereby it is crucial to unravel the genes associated with stress tolerance in this plant. The availability of a high-quality *S. miltiorrhiza* genome makes it possible to systematically analyze the *SmHsf* gene family [29]. In this study, a total of 35 *SmHsf* genes were identified from *S. miltiorrhiza*. The chromosomal distribution, phylogenetic relationship, gene structure, conserved domains, evolutionary relationships, and cis-regulatory elements were analyzed. We further explored the expression patterns across *SmHsf* genes using RNAseq data from different organs and various stresses. Finally, we investigated *SmHsf1*- and *SmHsf7*-mediated heat response in yeast. Our results provide a foundation for further functional research on *Hsf* genes in Danshen plants that will help the breeding and genetic engineering of this plant.

## 2. Results

### 2.1. Identification and Chromosomal Distribution of Hsf Genes in S. miltiorrhiza

We identified 35 non-redundant *SmHsf* genes in the *S. miltiorrhiza* genome, which we termed *SmHsf1* to *SmHsf35* according to their respective location along the chromosomes (Figure 1). Two genes (*SmHsf34* and *SmHsf35*) were located on unanchored scaffolds, while the remaining 33 were distributed on eight chromosomes.

Sequence analysis showed that most (77%) *SmHsfs* contained two exons, with coding sequences (CDS) varying from 558 bp (*SmHsf22*) to 1527 bp (*SmHsf9*) and a protein length ranging from 207 to 665 amino acids (Appendix A). The relative molecular weight (MW) of SmHsfs was 21.54 kDa (SmHsf22)–56.36 kDa (SmHsf9), the computed theoretical isoelectric points (pI) ranged from 4.64 (SmHsf23) to 9.93 (SmHsf27), and the instability index indicated that all SmHsf proteins were unstable. The grand average of hydropathicity (GRAVY) varied between −0.91 (SmHsf4) and −0.48 (SmHsf6). Additionally, subcellular localization analysis showed that all SmHsfs were localized in the nucleus, with the exception of SmHsfs 11, 18 and 30, which were also located in the cytoplasm and mitochondria.

### 2.2. Phylogenetic and Classification Analyses of SmHsfs

To explore the evolutionary relationships and classification of Hsfs, we built an unrooted phylogenetic tree consisting of 35, 25 and 22 Hsf proteins from *S. miltiorrhiza*, *Oryza sativa* and *Arabidopsis thaliana*, respectively (Figure 2). This allowed us to classify the SmHsf family members into three main groups, SmHsfA, SmHsfB, and SmHsfC, of which SmHsfA was the largest (62.8% of the total) with a total of 22 members distributed across nine subgroups (A1–A9). SmHsfB contained five subclasses (B1–B5) and 11 genes, representing 31.4% of the total, while SmHsfC represented 5.8% of the total and contained only two genes (*SmHsf28*, *SmHsf29*). As expected, the Hsfs of *S. miltiorrhiza* are phylogenetically closer to *A. thaliana* than *O. sativa*.

### 2.3. Gene Structure, Conserved Domains and Motif Analyses

To explore the structural diversity of the *SmHsf* gene family, we analyzed the intron–exon boundaries of the 35 identified *SmHsf* genes and the conservation of protein sequences. We found that 27 *SmHsf* genes contained two exons, with *SmHsf1*, *SmHsf7*, *SmHsf9*, *SmHsf11* and *SmHsf35* containing three exons each; *SmHsf4* and *SmHsf8* containing four exons; and *SmHsf22* containing five exons (Figure 3A). The exon–intron organization of *SmHsfs* suggested conservation during *S. miltiorrhiza* evolution. We predicted six conserved domains, including DBD, HR-A/B, NLS, NES, AHA, and RD, in the SmHsf family (Appendix A). Of these, DBD and HR-A/B were conserved across all SmHsf members. The DBD domain consisted of three α-helices and four β-sheets (Figure 3B and Appendix A), but SmHsf11 contained no β1β2 sheet and SmHsf22 had no α1α2 helices. While most SmHsfs contained both the NLS and NES domains indispensable for maintaining SmHsf homeostasis between the nucleus and cytoplasm, the AHA and RD domains were specific to each group. Specifically, AHA motifs were only detected in Group A (A1, A4, A5 and A7), while RD motifs were found in the C-terminus of Group B and were characterized by the presence of a tetrapeptide motif (LFGV) that represses transcription.

We then scanned 10 distinct motifs in the MEME server to identify sequence features. As shown in Figure 3C and Appendix A, all SmHsf members contained DBD conserved motifs 1, 2, and 3, except for SmHsf11, which contained motifs 2 and 3 but no motif 1; and SmHsf22, which contained motif 2 but not motifs 1 and 3. HR-A/B conserved motif 4 was also found in all SmHsf proteins. Finally, we identified several group-specific motifs, including motifs 5, 6 and 7, in Group A, motif 10 in the A6 subfamily, and motif 9 in the B4 subfamily. Similar motif composition was usually observed on the same SmHsf clade, indicating these genes possess similar regulatory functions in *S. miltiorrhiza*.

### 2.4. Gene Duplication and Synteny Analyses of Hsf Genes in S. miltiorrhiza

Gene duplication events are important drivers of functional differentiation and gene amplification [30]. We identified 15 paralogous pairs involving 21 *SmHsf* genes (60%) that resulted from whole-genome/segmental duplication events (Figure 4). We also found 14 genes that underwent dispersed duplication events (40%) (Appendix A), suggesting this process was important for the expansion of *SmHsf* genes.

To further explore the evolution patterns of the *SmHsf* family, we performed synteny analysis between *S. miltiorrhiza* and four dicots (*S. bowleyana*, *S. baicalensis*, *A. hispanicum*, and *A. thaliana*) and two monocots (*O. sativa* and *Zea mays*) (Figure 5). We identified syntenic relationships in *A. hispanicum* (28 genes), *S. bowleyana* (25 genes), *S. baicalensis* (21 genes), *A. thaliana* (19 genes), *O. sativa* (nine genes) and *Z. mays* (nine genes); and orthologous pairs between *S. miltiorrhiza* and these six species, numbering 42, 41, 30, 24, 24 and 17, respectively (Appendix A). A total of 13 *Hsf* collinear gene pairs were identified between *S. miltiorrhiza* and *S. bowleyana* and anchored to highly conserved syntenic blocks spanning > 100 genes. Moreover, *SmHsf7*, *SmHsf24* and *SmHsf32* were associated with three or four syntenic gene pairs between *S. miltiorrhiza* and *S. bowleyana*; while other genes showed no fewer than two collinear gene pairs between *S. miltiorrhiza* and other five species. Interestingly, *SmHsf1* and *SmHsf7* were identified in *S. miltiorrhiza* and all six representative species, indicating an ancestral origin for these orthologous pairs.

### 2.5. Analysis of Cis-Acting Elements and Upstream Regulators of SmHsfs

In order to understand the regulatory network associated with *SmHsfs* roles, the *cis*-elements located 2 kb upstream of each of the 35 *SmHsfs* were analyzed by plantCARE. Beyond core promoter elements, we discovered 25 types of cis-elements in the promoter region of *SmHsf* genes, including eight hormone-, 13 stress- and four development- and metabolism-associated elements (Figure 6A; Appendix A). Light-responsive elements were abundant, especially the ACE, Sp1, and MRE elements (Figure 6B). We also detected other stress-responsive cis-elements, such as MBS (drought-inducibility), LTR (low-temperature responsiveness), and the GC-motif (anoxic specific inducibility). Among the hormone-responsive elements, the most numerous was the ABA-related ABRE, followed by the TGACG-motif (MeJA-responsiveness), and the TCA-element (SA-responsiveness) (Figure 6B,C). Among the elements related to plant development and metabolism, 16 *SmHsfs* contained an O2-site (zein metabolism regulation), nine *SmHsfs* contained a GCN4_motif (endosperm expression), and four included elements associated with the regulation of the circadian rhythm. Intriguingly, only *SmHsf30* contained one heat shock element (heat response) in its promoter region (Appendix A).

The abundance of cis-elements in the promoter regions of *SmHsf* genes may be related to the regulation of signaling pathways involved in plant development and stress response. To test this hypothesis, we predicted the potential regulatory interactions between *SmHsfs* and their upstream regulators using *Arabidopsis* as a model (Figure 7A; Appendix A). A total of 521 transcript factors (TFs) were scanned and a GO enrichment analysis performed, which indicated that most TFs were associated with positive regulation of transcription, plant development and ethylene responsiveness (Figure 7B). The genes *SmHsf3*, *SmHsf29*, *SmHsf31*, *SmHsf32* and *SmHsf35* were regulated by a higher number of TFs (Figure 7C). The most abundant TFs were *AtTCP17*, *AtTCP21*, *AtNAC079*, *AtERF115* and *AtWRKY26* (Figure 7D).

### 2.6. Expression Patterns of SmHsfs in S. miltiorrhiza Organs

We analyzed the expression patterns of *SmHsf* genes in four organs of *S. miltiorrhiza* using transcriptomic data (SRP327565), and found most of the genes (23/35) were upregulated in roots—eight of these genes had an exclusive high expression in this organ (Figure 8A; Appendix A). In addition, we found higher expression levels of *SmHsf3*, *SmHsf11*, *SmHsf19* in stems; *SmHsf4*, *SmHsf21* in leaves; and *SmHsf8*, *SmHsf15* in flowers. Along with the aforementioned tissue specificity of gene expression, we found a high diversity of transcript abundance. For example, the *SmHsf20*, *SmHsf29*, *SmHsf32* genes were highly expressed in the root and leaves; *SmHsf2*, *SmHsf13*, *SmHsf30* in the root and stems; and *SmHsf35*, *SmHsf33*, *SmHsf1* in the root and flowers (Figure 8B). Notably, *SmHsf23* had a relatively high expression profile in the root, stems, and flowers.

### 2.7. Expression Patterns of SmHsfs in Response to Hormone Treatment

We then examined the expression patterns of *SmHsf* genes as a response to ABA, MeJA, SA, and GA based on transcriptomic data (Figure 9; Appendix A). Genes with TPM < 1 were excluded, while those with a fold-change in TPM > 1.5-fold compared to controls were considered significantly regulated. After exposure to ABA, the expression levels of eight *SmHsfs* (*SmHsf12*, *SmHsf25*, *SmHsf2*, *SmHsf1*, *SmHsf5*, *SmHsf7*, *SmHsf32*, *SmHsf29*) showed significant upregulation; while *SmHsf14*, *SmHsf30*, *SmHsf19* and *SmHsf18* were significantly downregulated (Figure 9A). Under GA stress, most of *SmHsfs* (60%) displayed a decrease in expression levels, especially *SmHsf2* and *SmHsf25* (Figure 9B). After MeJA treatment, the transcription levels of *SmHsf20*, *SmHsf31*, *SmHsf32* and *SmHsf33* increased significantly in early stages (1 h), while *SmHsf13*, *SmHsf18* and *SmHsf27* were expressed at high levels in later stages (6 h) (Figure 9C). We also found that the expression of *SmHsf31* and *SmHsf34* under SA stress was 18 and 12 times lower than controls, respectively (Figure 9D). Finally, some genes responded to more than one treatment, including the upregulation of *SmHsf1*, *SmHsf2* and *SmHsf25* by ABA and SA, and *SmHsf33* by SA and MeJA.

### 2.8. Expression Patterns of SmHsfs in Response to Various Abiotic Stresses

Abiotic stresses (heat stress, drought and UV) affect plant growth and development, constituting the major factor limiting crop production. After UV treatment, we identified 22 upregulated (81%), and five downregulated (*SmHsf3*, *SmHsf16*, *SmHsf19*, *SmHsf27*, *SmHsf34*) genes (Figure 10A). Under heat and drought stress, a high proportion of Group A *SmHsfs* showed low transcript abundance, while most members of Group B exhibited high expression levels (Figure 10B), especially *SmHsf1* and *SmHsf7* (8.7 and 6.1 times, respectively, the levels found in the control group). In addition, the genes *SmHsf5*, *SmHsf24* and *SmHsf33* were highly expressed following drought treatment (Figure 10C). Finally, we identified *SmHsfs* responding to multiple stresses, including significant upregulation of *SmHsf24* by heat and drought; *SmHsf1*, *SmHsf7*, *SmHsf4* and *SmHsf35* by heat and UV; and *SmHsf5* and *SmHsf33* by heat, drought and UV (Appendix A).

### 2.9. Overexpression of SmHsf1 and SmHsf7 Enhanced Thermotolerance in Yeast Recombinant Cells

Interspecific synteny analysis showed that *SmHsf1* and *SmHsf7* were conserved between dicots and monocots, indicating that both genes might play an important role in adaptation during evolution. Since the two genes were also among the top upregulated genes in response to heat stress, we hypothesized they may play a vital role in thermotolerance in *S. miltiorrhiza*. Accordingly, we started by performing subcellular localization analysis. As shown in Figure 11A, the control GFP signal was observed throughout the entire cell, whereas *SmHsf1*-GFP and *SmHsf7*-GFP were only expressed in the nucleus, suggesting these are nuclear proteins.

Since heat stress response is conserved in eukaryotes, we performed thermotolerance analysis in yeast [31]. To achieve this, we evaluated the growth of yeast cultures transfected with a pPIC3.5K expression vector and the vector containing *SmHsf1* and *SmHsf7* under normal and heat stress conditions. Both sets of transformed yeast cells grew equally at 30 °C (Figure 11B). However, when exposed to 50 °C, *SmHsf1/7*-transformed yeast cells outgrew control cells, demonstrating that heterologous expression of *SmHsf1* and *SmHsf7* improves thermotolerance in yeast.

## 3. Materials and Methods

### 3.1. Identification and Sequence Analysis of Hsf Genes from S. miltiorrhiza

The whole genome sequences and annotation files of *S. miltiorrhiza* (Accession No. GWHAOSJ00000000), *Scutellaria baicalensis* (GWHBJEC00000000), *Antirrhinum hispanicum* (GWHBFSA00000000), *Salvia bowleyana* (GWHASIU00000000) were obtained from the China National Center for Bioinformation (CNCB) database (https://ngdc.cncb.ac.cn/, accessed on 1 September 2022). The protein sequences of AtHsf and OsHsf were downloaded from TAIR (https://www.arabidopsis.org, accessed on 28 August 2022) and Rice data (https://www.ricedata.cn/gene/, accessed on 9 September 2022), respectively. Hidden Markov model (HMM) search and BLAST-P were combined to identify potential SmHsfs. Firstly, the conserved Hsf DNA-binding domain (PF00447) was used as a query to identify SmHSF proteins using the HMMER (v3.3) software with an e-value < 10^−5^. Secondly, AtHsfs and OsHsfs were queried to search for possible SmHsfs by Blast Several Sequences to a Big Database program with default parameters in TBtools [32]. Finally, all obtained SmHsf proteins were submitted to the SMART (http://smart.embl.de/smart/set_mode.cgi?NORMAL=1, accessed on 13 September 2022) and MARCOIL databases (http://toolkit.tuebingen.mpg.de/marcoil, accessed on 9 October 2022) to verify the presence of Hsf-type DBD domain and HR-A/B domains. After the removal of the same genes, the remaining genes were identified as *SmHsf* genes.

The physicochemical parameters of SmHsf proteins, including their theoretical molecular weight (MW), theoretical isoelectric point (*pI*), were analyzed using the ExPASy (https://web.expasy.org/protparam/, accessed on 20 October 2022). The protein subcellular localization was predicted using Plant-mPLoc (http://www.csbio.sjtu.edu.cn/bioinf/plant-multi/, accessed on 20 October 2022). NES domains were analyzed using Wregex server (http://wregex.ehubio.es, accessed on 20 October 2022). NLS domains was predicted using NLS Mapper (https://nls-mapper.iab.keio.ac.jp/cgi-bin/NLS_Mapper_form.cgi accessed on 20 October 2022). The MEME (https://meme-suite.org/meme/, accessed on 13 September 2022) was used to analyze SmHsf conserved motifs with the number of motifs = 10. Visualization of SmHsf motifs and the structures of *SmHsf* genes was performed using TBtools software. Phylogenetic trees were constructed using the neighbour-joining (NJ) method in MEGA (version 11) software with 1000 bootstrap replicates. Furthermore, the 2000 bp promoter regions upstream of the initiation codon (ATG) of *SmHsf* genes were searched in the PlantCARE database for identification of cis-regulatory elements, and the number and location of cis-elements in each *SmHsf* were visualized using Heatmap and Simple BioSequence viewer in the TBtools program.

### 3.2. Duplication and Synteny Analysis of Hsf Genes

The duplication types of *SmHsf* genes were examined by the Collinearity Scan toolkit software (MCScanX) with the default parameters [33]. To explore the syntenic relationships of the *Hsf* genes obtained from *S. miltiorrhiza* and other selected species, syntenic analysis were performed using MCScanX and visualization was completed by Advanced Circos and Dual Systeny Plot program in TBtools.

### 3.3. Plant Growth Conditions and Plant Stress Treatments

*S. miltiorrhiza* seedlings were cultured in a greenhouse at 25 °C, a relative humidity of 50–60%, and a 16 h light/8 h dark photoperiod. The uniform four-week-old plants transferred from soil were grown adaptively in 1/2 Hogland nutrient solution for one week. For abiotic stresses, seedings were treated with 15% PEG 6000, 42 °C, abscisic acid (ABA, 100 μM) and salicylic acid (SA, 100 μM) solution described previously [34]. The roots of samples were collected at 6 h; the untreated plants were used as controls (CK). Three biological replicates were prepared for each treatment. All of the harvested samples were frozen in liquid nitrogen and stored at −80 °C for RNA-seq.

### 3.4. Transcriptome Analysis of SmHsfs Expression Patterns in Different Organs and under Abiotic Treatments

We analyzed *SmHsf* expression patterns based on RNA-seq data. Raw data from four organs (root, stem, leaf and flower) in *S. miltiorrhiza* were download from the National Center for Biotechnology Information (NCBI) (https://www.ncbi.nlm.nih.gov/sra/?term=, accessed on 3 October 2022) under BioProject SRP327565. The public RNA-seq data of *S. miltiorrhiza* response to abiotic stresses were also obtained from NCBI included gibberellin (GA) (SRP282755), Methyl jasmonate (MeJA) (SRP111399) and ultraviolet radiation (UV) (SRP214513). Besides public data, we also constructed 15 libraries for transcriptome sequencing, which were performed by Illumina Novaseq platform (Beijing) with 150 bp paired-end reads including CK, ABA, SA, PEG and heat samples (PRJNA916388). After quality control of raw data, clean data were mapped to the reference transcriptome sequence by kallisto, and transcript abundances, indicated as TPM (transcripts per million) values, were calculated [35]. The expression profiles of each *SmHsf* were visualized using Fancy Heatmap Browser program and Heatmap program in TBtools.

### 3.5. Subcellular Localization

The CDS of *SmHsf1* and *SmHsf1* without the stop codon were cloned in-frame into the *pCAMBIA1300-C-GFP* plant expression vector using a Basic Seamless Cloning and Assembly Kit (TransGen Biotech, Beijing). The constructed plasmids of *pCAMBIA1300-SmHsf1/7-GFP* and empty vector were transformed into *Agrobacterium tumefaciens* strain GV3101 Chemically Competent Cell (Coolaber, Beijing, China) by the conventional freezing thawing method. Next, all vectors GV3101 were transformed into 4-week-old *Nicotiana benthamiana* leaves with incubation buffer (10 mM MES pH 5.6, 10 mM MgCl_2_, 150 µM acetosyringone). In addition, *HY5-mcherry* was used as the nucleus-located marker. Fluorescent images were obtained at 48–60 h after infiltration by confocal laser microscopy (Zeiss LSM880).

### 3.6. Thermotolerance Analysis of Transgenic Yeast

The complete *SmHsf1* and *SmHsf7* (Open Reading Frame) ORF were inserted into the yeast expression vector *pPIC3.5K* (Coolaber). The plasmid *pPIC3.5K*-*SmHsf1/7* and *pPIC3.5K* were introduced into *Pichia pastoris* strain SMD1168 (Coolaber) using the lithium acetate method. A single positive clone of SMD1168 harboring the recombinant *pPIC3.5K*-*SmHsf1/7* or *pPIC3.5K* vector was incubated into 5 mL SD/-His liquid medium to grow overnight at 30 °C with 200 rpm. The overnight culture was diluted with sterile water to OD600 = 0.8. In addition, then one group of diluted culture were incubated in a water bath at 50 °C for 20 min, another group at 30 °C for 20 min. After heat treatment, 10 μL yeast cells of 10-fold serial dilutions were dotted on SD/-His plates. All plate were photographed after four days of incubation of 30 °C.

## 4. Discussion

Because they are sessile, plants have evolved complicated transcription control systems to adapt to fluctuating environmental conditions, and a growing amount of evidence suggests Hsfs are essential regulators of this process [19,36]. Genome-wide analysis of the Hsf family were performed in *Arabidopsis* [36], rice [36], maize [37], Chinese cabbage [38], and so on. However, the Hsf family has not been extensively studied in *S. miltiorrhiza*.

Here, we identified 35 *Hsf* genes from the *S. miltiorrhiza* genome, which were classified into three groups based on evolutionary relationships and respective characteristics. When compared to *Arabidopsis* and rice, *S. miltiorrhiza* contains a higher number of SmHsfs [36], with each subgroup showing partial expansion (except subgroupA1) as a result of gene duplication events. In particular, subgroup A9 contained six members in *S. miltiorrhiza*, but only one in *Arabidopsis* [36]. The reasons behind this expansion require further investigation, but are usually related to novel traits [39]. We also found that *SmHsf* genes shared similar exon–intron patterns. Since introns can significantly affect gene expression [40], the similar structures and conserved motifs found within subgroups indicated they may have similar functions.

Gene families are groups of genes originating from a common ancestor, which often retain similarities [41]. Rapid gene family expansion of phenotypically important genes can help plants adapt to adverse environmental conditions [42]. Several gene duplication events contribute differentially to gene family expansion in plants, including WGD/segment duplications, tandem/local duplications, and dispersed duplications, of which the former two account for the majority of duplicates [43,44,45]. A total of three WGD events in *Arabidopsis* led to a 90% increase in regulatory genes [46]. As for *S. miltiorrhiza*, it experienced a very recent species-specific duplication event and multiple TE (transposable elements) transposition bursts [29]. In fact, intraspecific synteny analysis showed that the expansion of the smHsf family derived primarily from WGD or segmental duplications, which was also observed in maize [37]. We also found that TE-mediated dispersed duplications were the other major drivers of this gene family expansion.

DanShen is an important medicinal plant, whose breeding can be improved by first cultivating a few thin branching rootlets and screening the genes that control lateral root [27]. Previous studies have shown that Hsf TFs play important roles during root development in plants. For example, the *Athsfa4c* mutant has a reduced number of lateral and smaller roots [47], while root length and number of lateral roots were significantly lower in *Arabidopsis* overexpressing *Prunus persica HSF5* [48]. Accordingly, we explored the expression profiles of *SmHsfs* in four different organs to excavate the *SmHsf* genes that potentially regulate Danshen root development. While we observed tissue-specific expression patterns in *SmHsfs*, such as *SmHsf3* in stems, *SmHsf4* in leaves and *SmHsf15* in flowers, most genes were highly expressed in the root, including *SmHsf5*, *SmHsf7*, *SmHsf14* and *SmHsf26*. These genes may thus play a major role in root developmental processes in this plant.

The expression patterns of *SmHsfs* following stress induction may lead to a deeper understanding of their biological functions. We found that 11 *SmHsf* genes were induced by heat stress (42 °C), including significantly higher expressions of SmHsfA2, A7, B1, B2, B3 and B5, which can be major regulators of *Hsp*- or heat-inducible genes. A previous study showed that *AtHSFA2* is the most strongly induced *Hsf* gene under heat stress, with mutant lines displaying thermosensitivity [7]. Moreover, the expressions of *DcaHsf*-A2a and *A2b* were highly upregulated under heat stress in carnation plants [49]; the expression of *AtHSFA2*, *LlHSFA2* and *OsHSFA2e* enhanced thermotolerance in transgenic *Arabidopsis* [50,51]; and *HsfA7* expression in tomato rapidly increased under mild heat stress, after which the complex HsfA7-HsfA1a was formed to modulate heat stress response [4]. Considering these studies, it is possible that SmHsfA2 and SmHsfA7 may also enhance thermotolerance in *S. miltiorrhiza*.

Several reports have documented that the majority of HsfA members are upregulated by heat stress and play a particularly important role in thermotolerance [20,50]. However, our results showed that most of the subgroup SmHsfB genes (6/11) were significantly upregulated by heat stress. In addition, SmHsfB2 (*SmHsf1,7*) and SmHsfB3 (*SmHsf5*) contained the tetrapeptide expression repressor LFGV, which was not present in SmHsfB1 (*SmHsf33*), SmHsfB2 (*SmHsf24*) and SmHsfB5 (*SmHsf27*). The few studies available on groups HsfB and HsfC were vital for assessing heat-stress recovery in plants [7]. The repressive activity of *HsfB1* and *HsfB2* were confirmed in *Arabidopsis*, as these genes could negatively regulate the expression of heat-inducible *Hsfs,* such as *HsfA2* [52]. However, tomato HsfB1 is a coactivator factor that assembles with HsfA1 into an enhanceosome-like complex to synergistically activate a reporter gene [53].

We analyzed the biological function of the most highly induced *SmHsf1,7* genes under heat stress in yeast cells. Thermotolerance assays suggested that overexpression of *SmHsf1* and *SmHsf7* in yeast enhanced resistance to high temperatures, whereby it seems plausible that *SmHsf1* and *SmHsf7* act as coactivators and not as repressors. However, the exact mechanisms require further investigation. We also note that expression levels of SmHsfC (*SmHsf29*) were significantly decreased under heat stress, suggesting that the gene has a negative control function.

Previous work has demonstrated that plant Hsfs regulate various stress responses and overlap considerably [54]. In *Arabidopsis*, cold, salt, ABA, drought and UV-B stress activate the *HsfA2*, *HsfA4*, *HsfA6B*, *HSFB1*, *HSFB2A*, and *HSFB2B* genes to varying degrees [9]. The ectopic expression of *Hsfs* demonstrates it is possible to enhance *Arabidopsis* or tobacco tolerance to various stresses, including high salinity, osmotic, extreme temperatures, or UV light [55]. According to the cues of cis-acting elements analysis, the expression patterns of *SmHsfs* were comprehensively analyzed under treatments of ABA, MeJA, SA, GA, PEG and UV. Most of these genes responded to more than one stress—upregulation or downregulation, which can point out directions of further investigations of biological function.

## 5. Conclusions

We identified 35 *SmHsfs* in the *S. miltiorrhiza* genome and performed a comprehensive bioinformatic analysis on the newly characterized. Firstly, we examined the chromosome distribution, gene structures, conserved domains, gene duplication and synteny analysis. After this, cis-element analysis allowed us to specifically investigate the expression profiles of *SmHsfs* in four different organs, as well as their expression profiles in response to multiple abiotic stresses and phytohormones. Furthermore, our results revealed that *SmHsf1* and *SmHsf7* belonged to SmHsfB and perform similar heat response functions in yeast. Yet more studies are needed to determine the precise pathway linking the *Hsf* gene to heat stress in *S. miltiorrhiza*. This research provides a basis for the functional study of *SmHsfs* and mechanisms of resistance to environmental change in *S. miltiorrhiza*.

## Figures and Tables

**Figure 1 ijms-24-08461-f001:**
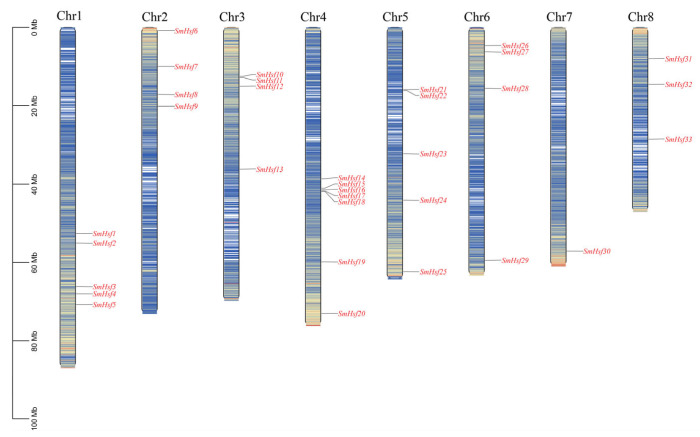
Chromosomal locations of *Hsf* in *S. miltiorrhiza* on chromosomes 1–8. The scale represents sequence length in megabases (Mb).

**Figure 2 ijms-24-08461-f002:**
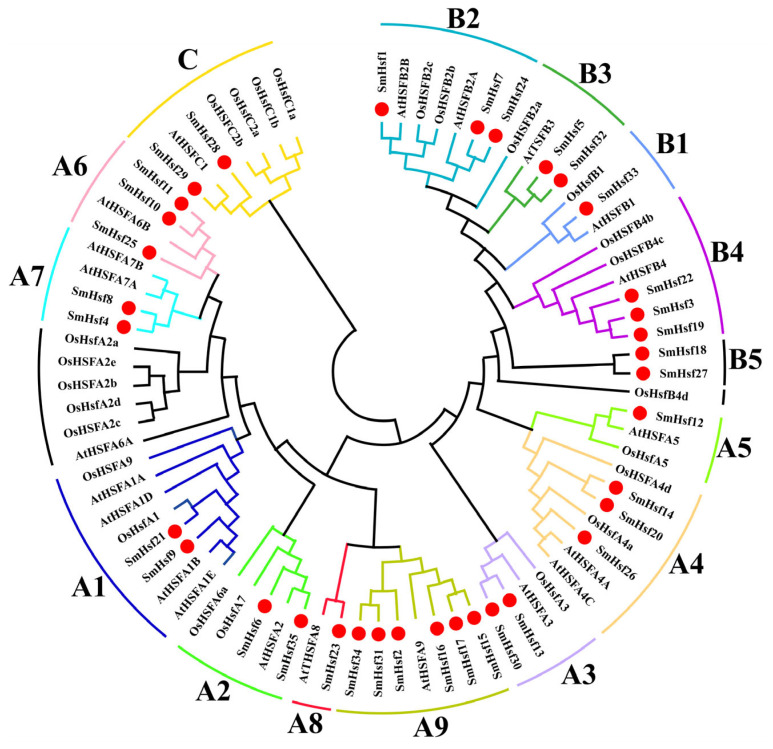
Phylogenetic tree of the *Hsf* gene family from *S. miltiorrhiza*, *A. thaliana*, and *O. Sativa*. The phylogenetic tree was constructed using MEGA (version 11) with the neighbor-joining (NJ) method (1000 bootstrap replicates). *Hsfs* in *S. miltiorrhiza* are indicated by the red nodes.

**Figure 3 ijms-24-08461-f003:**
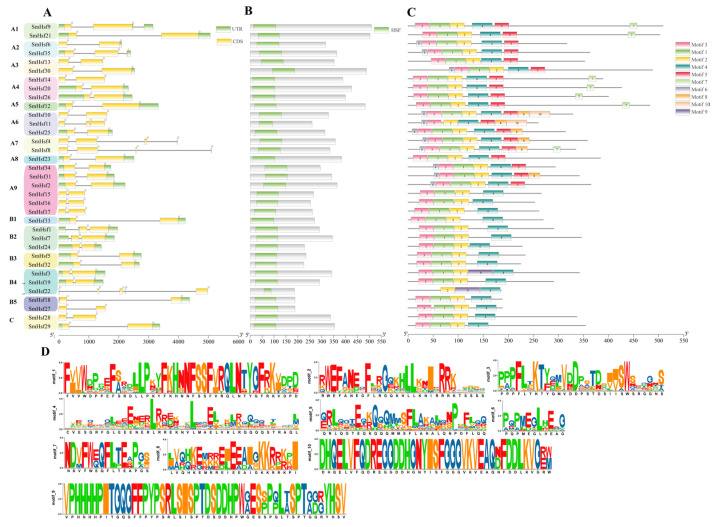
Gene structure, conserved domains and motif analysis of SmHsf proteins from *S. miltiorrhiza*. (**A**) Exon–intron structures of *SmHsf*. UTRs, exons and introns were represented by green boxes, yellow boxes and black single lines, respectively. (**B**) The conserved domain in the SmHsf proteins. Green boxes indicate Hsf or DBD domain. (**C**) Distribution of conserved motifs of the SmHsf proteins. The ten motifs are represented by different colored boxes, of which sequence logos were showed in (**D**). The sequence length of the domain or motif was indicated at the bottom of each figure.

**Figure 4 ijms-24-08461-f004:**
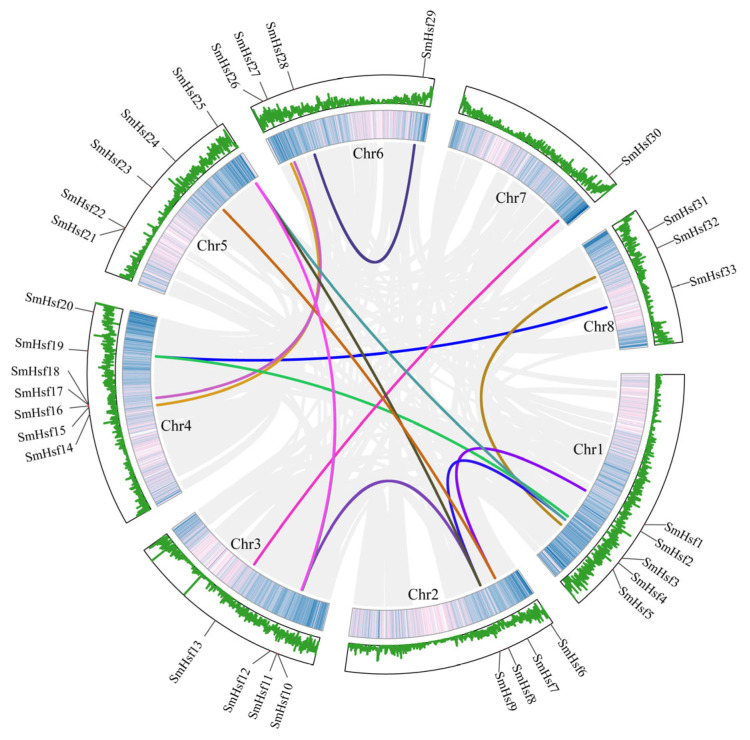
The inter-chromosomal relationships of *SmHsf* genes. The two outer rings represent the gene density per chromosome of *S. miltiorrhiza*. Gray lines represent synteny blocks in the genome, while duplicated *SmHsf* gene pairs are connected with different color lines.

**Figure 5 ijms-24-08461-f005:**
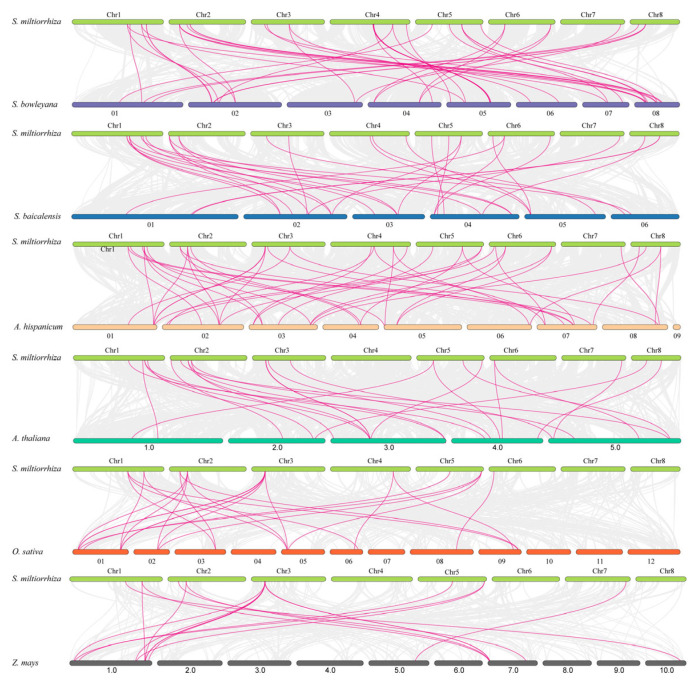
Synteny analysis of *Hsfs* between *S. miltiorrhiza* and six other species. The collinear blocks are marked by gray lines, while the collinear gene pairs with *Hsf* genes are highlighted in the purple lines. The different color bar indicated the different chromosomes. ‘*S. bowleyana*’, ‘*S. baicalensis*’, ‘*A. hispanicum*’, ‘*A. thaliana*’, ‘*O. sativa*’ and ‘*Z. mays*’ indicate *Salvia bowleyana*, *Scutellaria baicalensis*, *Antirrhinum hispanicum*, *Arabidopsis thaliana*, *Oryza sativa* and *Zea mays*, respectively.

**Figure 6 ijms-24-08461-f006:**
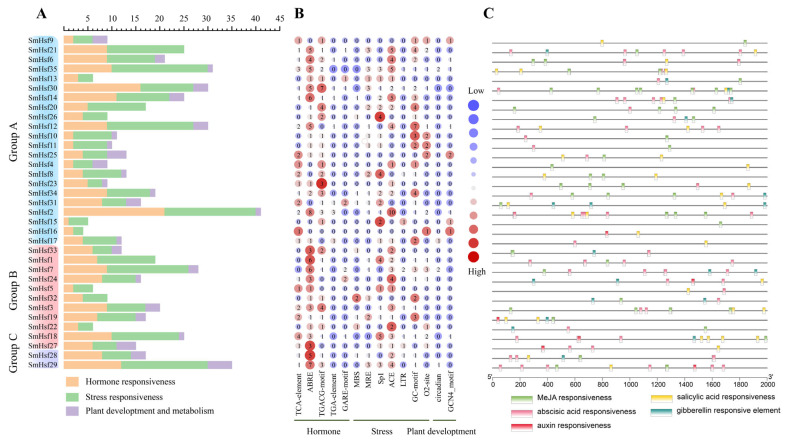
Analysis of the cis-elements in *SmHsf* gene promoters. (**A**) The sum of the *cis*-elements in each category is demonstrated by the varied color bar diagram. (**B**) Heatmap of the numbers of major cis-elements in hormone-responsive, stress responsiveness and plant development and metabolism. (**C**) Distribution of hormone-responsive elements in the promoter region, including position, kind and quantity of elements.

**Figure 7 ijms-24-08461-f007:**
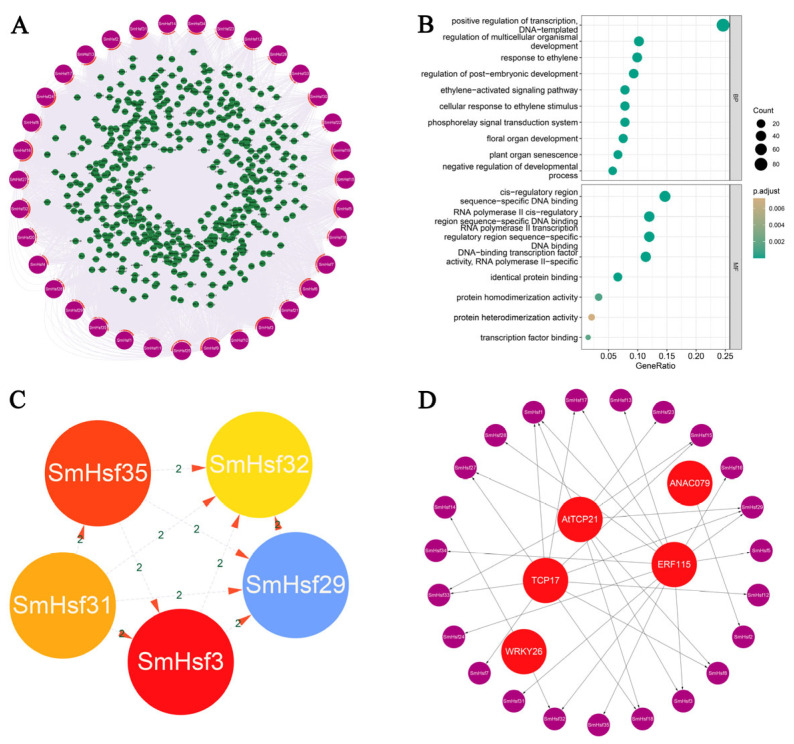
The prediction of the regulation network between *SmHsfs* and upstream regulators. (**A**) Construction of the network using Cytoscape. The purple box represents 35 *SmHsf* genes, the green boxes represent different upstream regulators. (**B**) GO enrichment analysis of all upstream regulators in the network. (**C**) The top five regulated *SmHsfs.* (**D**) The top five upstream regulators.

**Figure 8 ijms-24-08461-f008:**
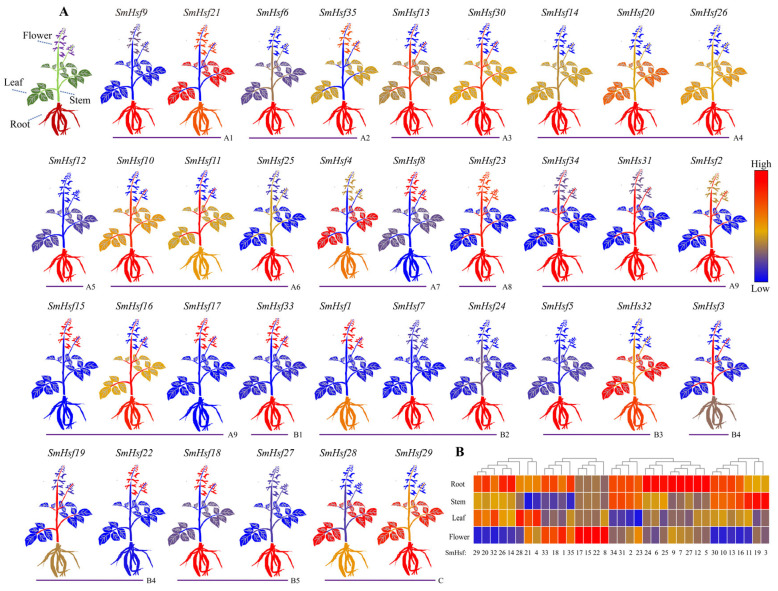
Expression profile of *SmHsf* genes in different organs. The investigation based on a public transcriptome data of *S. miltiorrhiza* (SRP327565), and the mean expression value of three biological replications was used for each organ sample. (**A**) Diagram showing the different organs of the 2-year-old *S. miltiorrhiza*, including root, stem, leaf and flower. (**B**) The heatmaps were generated using TBtools. Red color signifies high expression level, while blue signifies low expression level.

**Figure 9 ijms-24-08461-f009:**
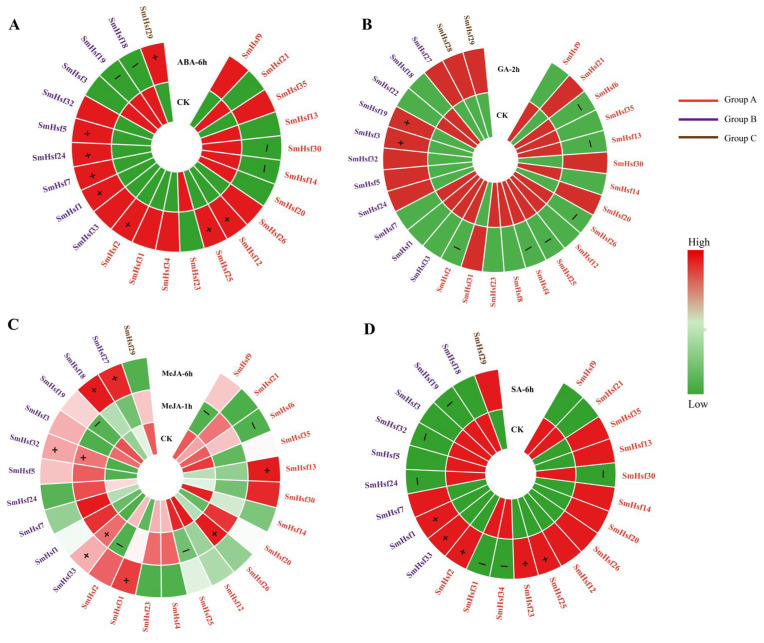
Expression profiles of 35 *SmHsf* genes in roots of *S. miltiorrhiza* under exogenous phytohormone treatments. The transcriptional levels of *SmHsf* genes in response to ABA (**A**), GA (**B**), MeJA (**C**) and SA (**D**) treatments were analyzed based on transcriptome data. The normalized TPM values indicated expression level of genes, which green indicates downregulation, and red represents upregulation. The genes with fold-change > 1.5 were considered as significantly changed genes. The significantly upregulated genes are marked by “+” and significantly downregulated genes are marked by “−”. *SmHsfA*, *SmHsfB* and *SmHsfC* are grouped with red, purple and brown colors. CK, control; ABA-6h, abscisic acid treatment for 6 h; GA-2 h, gibberellin treatment for 2 h; MeJA-1 h/6 h, methyl jasmonate treatment for 1 h or 6 h; SA-6 h, salicylic acid treatment for 6 h. Detail data in Appendix A.

**Figure 10 ijms-24-08461-f010:**
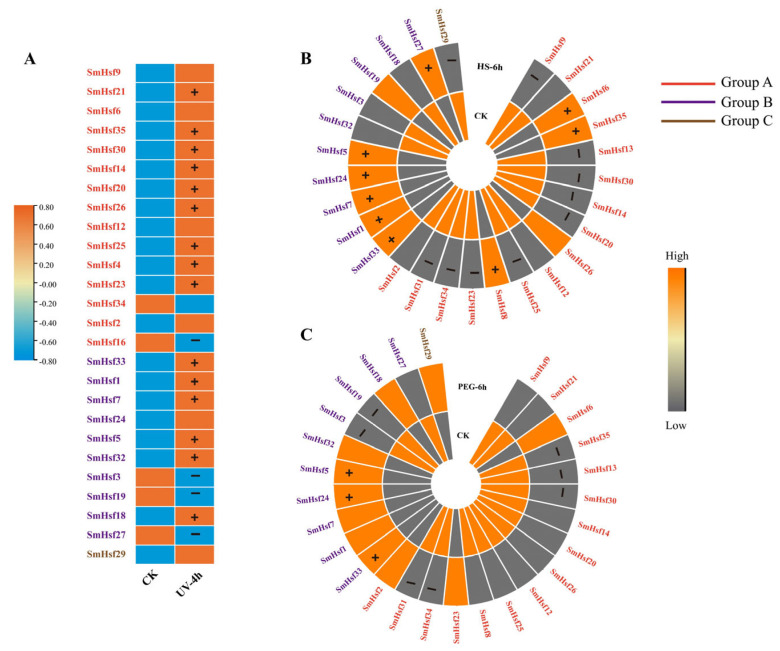
Expression patterns of 35 *SmHsf* genes in roots of *S. miltiorrhiza* under various abiotic stresses. The transcriptional levels of *SmHsf* genes in response to UV−light (**A**), heat stress (HS) (**B**) and PEG (**C**) stressors were analyzed based on transcriptome data. The normalized TPM values indicated expression level of genes. The genes with fold−change > 1.5 were considered as significantly changed genes. The significantly upregulated genes are marked by “+” and significantly downregulated genes are marked by “−”. *SmHsfA*, *SmHsfB* and *SmHsfC* are grouped with red, purple and brown colors. CK, control; UV−4 h, UV−light treatment for 4 h; HS−6 h, 42 °C treatment for 6 h; PEG−6 h, PEG−6000 treatment for 6 h. Detail data in Appendix A.

**Figure 11 ijms-24-08461-f011:**
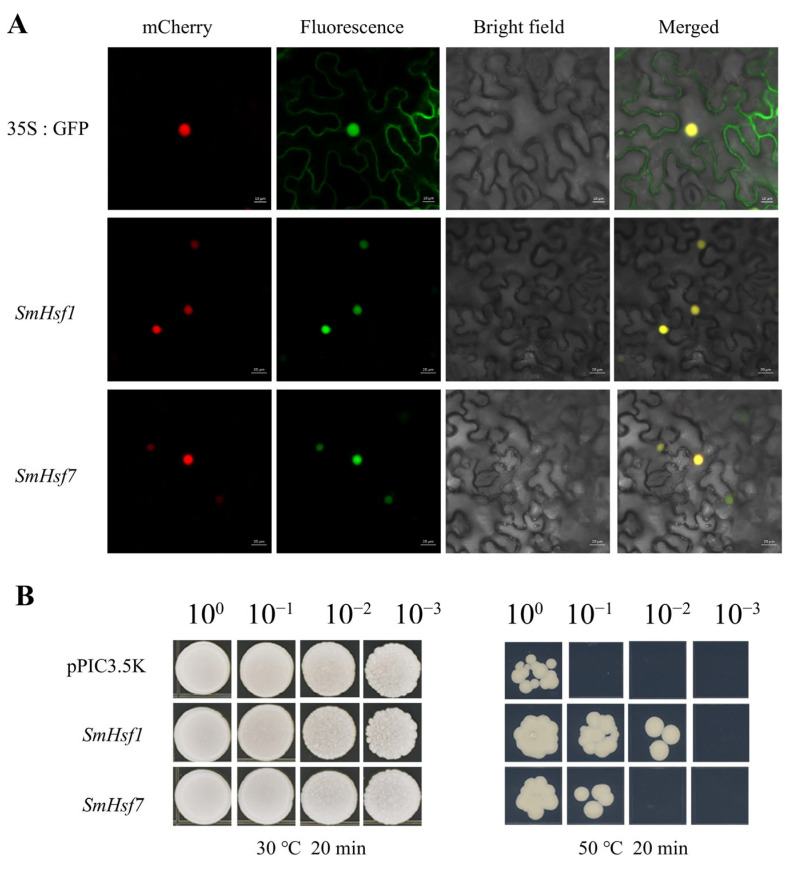
Characteristic analysis of *SmHsf1* and *SmHsf7*. (**A**) Subcellular localization analysis in tobacco epidermal cells. HY5-mcherry was used as indication of nuclear localization. (**B**) Growth of *P. pastoris* SMD1168 transformants after HS (50 °C) and normal temperature (30 °C). pPIC3.5K, SMD1168 cells carrying empty vector; *SmHsf1/7*, SMD1168 cells carrying vector pPIC3.5k-*SmHsf1/7*.

## Data Availability

Data generated in this work is provided in the manuscript.

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
