# Peer review of "Genome-Wide Characterization and Expression of the Hsf Gene Family in Salvia miltiorrhiza (Danshen) and the Potential Thermotolerance of SmHsf1 and SmHsf7 in Yeast"

_ijms, 2023, doi:10.3390/ijms24108461_

Round 1

Reviewer 1 Report

The MS entitled "Genome-wide characterization of the Hsf gene family in Salvia 1 miltiorrhiza (Danshen) and role of SmHsfB2 in heat tolerance' is well structured, having interesting information. However, before accepting for the publication the author need to revise the current version of the manuscript as suggested below. Authors should address following queries carefully and should submit a rebuttal.

  1. Check and mention the scientific and common names in the whole manuscript. Please add scientific names at first mention and used common names consistently thereafter.
  2. Please check carefully the spacing between words in the whole text.
  3. Most of the abbreviations in the manuscript are not elaborated. Please re-check and elaborate them at first mention and used consistently thereafter.
  4. Arrange the key words alphabetically.
  5. Please be checked the whole manuscript, the format should be improved such as there are some unit presentations, put the relevant unit with each value across the manuscript.
  6. Revised the abstract to sustain the essence of the proposed title, since the title of this review is too broad but justification has not been doe accordingly.
  7. The introduction section is well written and in my opinion it sheds light on the problem in a concise manner, but it should be more elaborated. Authors are suggested to write a brief about the long-term significant contribution of impacts of heat stress on plant and soil health by stating some statistical approximation.
  8. Mention the implicational strategies associated with Hsf gene in agricultural field and food security in brief.
  9. Conclusion part is not satisfactory and can be modified. The prospects need to be rewritten because the profound relationship between heat stress and Hsf gene is not elucidated, so future research directions should be better pointed out.

Reviewer 2 Report

Title:

Genome-wide characterization of the Hsf gene family in Salvia miltiorrhiza (Danshen) and role of SmHsfB2 in heat tolerance

General comments:

The manuscript is well written and organized!

What statistical analysis was used?

Specific comments

Abstract

Salvia miltiorrhiza Bunge (Danshen) is a traditional Chinese herb with significant medicinal value.  12 The yield and quality of Danshen are greatly affected by climatic conditions, in particular high temperatures.  13 Heat shock  factors (Hsfs)  play important  regulatory  roles in  plant response to heat and other environmental  14 stresses. However,  little is  currently  known about  the role played by the  Hsf gene family in  S. miltiorrhiza.  15 Here,  we identified  35  SmHsf  genes  and  classified  them  into  three  major  groups  (SmHsfA, SmHsfB, and  16 SmHsfC) using phylogenetic analysis. The gene structure and protein motifs were relatively conserved within  17 subgroups  but diverged  among the  different groups.  The expansion of  the  SmHsf  gene family was mainly  18 driven by whole-genome/segmental and dispersed gene duplications.  The expression profile of SmHsfs in four  19 distinct organs  revealed  its  members are  predominantly expressed  in  the  root.  The expression of a large number  20 of  SmHsfs  was  regulated by drought, UV, heat  and  exogenous  hormones.  Notably,  the  SmHsf1 and SmHsf7  21 genes in SmHsfB2 were the most responsive to heat and are conserved between dicots and monocots. Finally,  22 heterologous expression analysis showed that  SmHsf1  and  SmHsf7  enhance thermotolerance in yeast.  Our  23 results  provide a solid  foundation  for further functional investigation of  SmHsfs  in Danshen  plants as a  re- 24 sponse to abiotic stresses.  25

Could you please show some data in the abstract?

Why the figures are not included in the text?

Round 2

Reviewer 2 Report

Thanks to the authors for applying my comments accordingly. All the best!